# Codon Pair Deoptimization (CPD)-Attenuated PRRSV-1 Vaccination Exhibit Immunity to Virulent PRRSV Challenge in Pigs

**DOI:** 10.3390/vaccines11040777

**Published:** 2023-03-31

**Authors:** Min-A Lee, Su-Hwa You, Usharani Jayaramaiah, Eun-Gyeong Shin, Seung-Min Song, Lanjeong Ju, Seok-Jin Kang, Sun Hee Cho, Bang-Hun Hyun, Hyang-Sim Lee

**Affiliations:** 1PRRS Research Laboratory, Viral Disease Division, Animal and Plant Quarantine Agency, Gimcheon 39660, Republic of Korea; 2Department of Animal Veterinary Development, BioPOA, 593-26 Dongtangiheung-ro, Hwaseong 18469, Republic of Korea

**Keywords:** porcine reproductive and respiratory syndrome virus type (PRRSV)-1, codon pair deoptimization (CPD), ORF7 gene, vaccine

## Abstract

Commercially used porcine respiratory and reproductive syndrome (PRRS) modified live virus (MLV) vaccines provide limited protection with heterologous viruses, can revert back to a virulent form and they tend to recombine with circulating wild-type strains. Codon pair deoptimization (CPD) is an advanced method to attenuate a virus that overcomes the disadvantages of MLV vaccines and is effective in various virus vaccine models. The CPD vaccine against PRRSV-2 was successfully tested in our previous study. The co-existence of PRRSV-1 and -2 in the same herd demands protective immunity against both viruses. In this study, live attenuated PRRSV-1 was constructed by recoding 22 base pairs in the ORF7 gene of the E38 strain. The efficacy and safety of the CPD live attenuated vaccine E38-ORF7 CPD to protect against virulent PRRSV-1 were evaluated. Viral load, and respiratory and lung lesion scores were significantly reduced in animals vaccinated with E38-ORF7 CPD. Vaccinated animals were seropositive by 14 days post-vaccination with an increased level of interferon-γ secreting cells. In conclusion, the codon-pair-deoptimized vaccine was easily attenuated and displayed protective immunity against virulent heterologous PRRSV-1.

## 1. Introduction

The global porcine industry is facing a major threat due to the devastating economic losses caused by porcine reproductive and respiratory syndrome virus (PRRSV) [1]. Reproductive failure in sows and respiratory disease in young growing pigs are the characteristic features of PRRSV infection [2,3]. PRRSV is a positive-sense RNA virus belonging to the order *Nidovirales*, family *Arteriviridae*. It has two antigenically distinct species, *Betaarterivirus suid 1* (PRRSV-1) and *Betaarterivirus suid 2* (PRRSV-2) [4,5]. Furthermore, there are four subtypes of PRRSV-1 and nine lineages of PRRSV-2 [6] based on ORF5 gene-based phylogenetic analysis. Genetic and antigenic diversity of PRRSV has increased over time, leading to the emergence of new genotypes [7].

A mystery swine disease hit the USA for the first time in 1987 and later spread across the world [8]. It was called various names and officially named porcine reproductive and respiratory syndrome (PRRS) [9]. PRRSV is roughly oval-to-spherical shaped and bears positive-sense RNA within an enveloped nucleocapsid. Positive-sense RNA of 15 kb encodes 11 ORFs and two untranslated regions. ORF1a and ORF1b encode nonstructural replicase polyproteins involved in replication of the viral genome, whereas ORF2–7 encode structural proteins such as glycoprotein GP2a, GP2b, GP3–GP5, envelope (E) protein, membrane (M) protein, and nucleocapsid (N) protein [10,11].

The N protein encoded by ORF7, a small basic heterodimeric serine phosphoprotein with a molecular mass of 12–15 kDa, is the most immunodominant structural protein in PRRSV-infected cells [12,13,14]. It is involved in several immune evasion mechanisms. The nuclear localization signal present in N protein helps it to reach the nucleolus of infected cells [15]. N protein induces interleukin (IL)-10 secretion from infected host cells [16] and is an interferon (IFN)-γ antagonist [10].

PRRSV displays complex interactions with the immune system and a high mutation rate. Biosecurity and vaccination with modified live virus (MLV) and inactivated virus vaccines are currently used to control the spread of PRRSV. Vaccination with MLV vaccines is a long-established practice in swine herds. However, these vaccines have several shortcomings such as a partial cross-protection against heterologous virus strains, a high mutation rate with reversion back to an infectious form in passaged cell lines and vaccinated animals, and a tendency to recombine with wild-type PRRSV strains and thereby generate new virus strains with high genetic diversity and complexity [9,17,18,19].

A codon pair deoptimization (CPD) attenuates a virus to decrease codon pair bias (CPB), which is the average score of codon pairs making up a gene. Usage of underrepresented synonymous codons rather than common specific codons attenuates a virus via introduction of several point mutations that alter the nucleotide triplicate while preserving the amino acid composition of the protein. The generated attenuated virus has immunological and replication patterns that are very similar to those of the wild-type virus. Several pinpoint mutations are introduced and therefore the modified virus cannot revert and is safe. Codon pair deoptimization (CPD) helps to controllably and predictably attenuate viruses [20,21].

Several viruses such as polio virus [20], human immunodeficiency virus [22], influenza virus [23], human respiratory syncytial virus [24], Zika virus [25], foot and mouth disease virus [26], and vaccinia virus [27] display attenuation upon CPD both in vitro and in vivo. CPD has been successfully applied to attenuate PRRSV by targeting NSP2, NSP9, ORF5, and NSP1 [11,28,29,30,31].

Until 2000, PRRSV-2 infection was predominant, but since then, the occurrence of the PRRSV-1 has been continuously increasing in Korea. Currently, both genotypes are circulating in farms [32]. An MLV vaccine that is efficacious against one type of PRRSV will not necessarily offer the same level of efficacy against a different type of PRRSV in the same country [9]. A PRRSV-2 vaccine gives only partial protection against PRRSV-1 [33,34,35] because these 2 viruses only share ~57% nucleotide identity. Hence, vaccination with commercial vaccines against each type of PRRSV generated using the domestic endemic viruses is wise to effectively control infection with both viruses. Currently, the subtype 1 (subgroup C)-based MLV vaccine is used in Korea, but subtype 1 (subgroup A) has mainly occurred on swine farms [36]. Codon-pair-deoptimized attenuated domestic vaccine against PRRSV-1 is needed to deal with the rapid increase in infection of this virus on Korean swine farms.

In this study, Korean field isolate was attenuated using CPD on ORF7. The efficacy of CPD on attenuation of the virulent strains was confirmed, and the evaluation on the induction, which was designed to clarify protective immunity against a virulent strain, was investigated.

## 2. Materials and Methods

### 2.1. Virus and Cells

The E38 strain isolated from a swine on a Korean farm infected with PRRSV-1 in 2007 (GenBank No: KT033457) was subjected to CPD. The virus belongs to subtype 1A of PRRSV-1 [32]. CBNU0495 (GenBank No: KY434183) was isolated from lung of PRRS-affected pig due to its high pathogenicity, and it was used as a challenge virus. The viral sequence analysis demonstrated that two viruses, E38 and CBNU0495 belonged to subtype 1 (subgroup A) and nucleotide similarity between E38 (strain used for attenuation) and CBNU0495 (experimental challenge virus) is 93%. MARC-145 cells were used to propagate, transfect, and rescue the viruses used in this study. PAM cells from the lung lavage of PRRSV-free pigs were used to evaluate viral attenuation by CPD. The cells were maintained in the appropriate medium as described previously [11].

### 2.2. Full Genome Sequencing of E38

RNA was extracted using a Viral Gene-spin Extraction Kit (iNtRON Biotechnology, Sungnam-si, Gyeonggi-do, Republic of Korea) to perform full genomic sequencing of E38. The full-length viral genome was amplified in fragments using a OneStep RT-PCR Kit (Qiagen, Hilden, Germany) with 24 primer pairs designed based on the sequence of E38 [34]. Untranslated regions (UTRs) at the 5′ and 3′ ends of the viral genome were amplified with the 5′ and 3′ RACE system following the manufacturer’s instructions (Invitrogen, Waltham, CA, USA). The amplified products were subjected to Sanger’s dideoxy sequencing (Macrogen, Seoul, Republic of Korea).

### 2.3. Reverse Genetics

#### 2.3.1. Development of a Computer Program Algorithm and CPD of E38

A computer program algorithm was developed to calculate CPB of the PRRSV genome upon CPD. A total of 44,564 genes were downloaded from NCBI (http://www.ncbi.nlm.nih.gov/genome/term=swine (accessed on 1 March 2019)) to calculate CPB in the swine genome. The codon pair deoptimization for the selected genome region was followed according to previous reports [11,20,28]. CPB was calculated based on the average codon pair scores in the selected genome region. Modification of ORF7 of E38 was initially designed for two versions with different levels of CPD. CPD was applied to the gene by a step-by-step approach, in which the nucleotides were sequentially altered from the previous version to generate the next version. Through shuffling the original nucleotide sequence, CPB of ORF7 of E38 decreased, while generating silent mutations without changing the original amino acid sequences. After generating the attenuated codon-pair-deoptimized virus (E38-ORF7 CPD) using the E38 strain, RNA structure prediction software (http://rna.tbi.univie.ac.at/cgi-bin/RNAfold.cgi (accessed on 1 March 2019)) was used to calculate the minimum free energy needed to maintain the secondary structure. In addition, dinucleotide frequencies were obtained using the program composition scan in the SSE package [37].

#### 2.3.2. DNA Synthesis and Assembly of Clones

For full-length cDNA clone assembly, six fragments representing full genome CPD of E38 were synthesized de novo (BIONEER Co., Daejeon, Republic of Korea). The last fragment contained the codon pair deoptimized or original region of ORF7 inserted at the *BsgI* and *PacI* restriction sites as shown in Figure 1.

The cytomegalovirus promoter and hammerhead ribozyme were inserted before the 5′ UTR. The hepatitis delta virus ribozyme sequence was added after the 3′ UTR of the viral genome.

Silent mutations were introduced by de novo synthesis to insert the restriction enzyme sites. All fragments were ligated using shared restriction sites, which were designed to produce two full-length PRRSV-1 cDNA clones (pUC-Kana-E38-CPD and pUC-Kana E38 original). The assembled full-length cDNA clones were confirmed by full genomic sequencing. An illustration of the clone is presented in the Figure 1. The resultant viruses were designated E38-ORF7 CPD (attenuated virus) and the E38 original virus (non-attenuated virus).

#### 2.3.3. Transfection and Rescue of Viruses

The E38-ORF7 CPD full-length cDNA clone was transfected into MARC-145 cells [11], and the cell culture supernatant was collected at 120 h post-transfection and used to infect MARC-145 cells. To confirm rescue of the virus, an immunofluorescence assay was performed using a PRRSV-specific anti-N protein antibody at 96 h post-infection (SR-30; Rural Technologies Inc., Brookings, SD, USA). The modified ORF7 gene from the rescued virus was amplified every five passages using specific primers (forward primer: GCCTTTAGCATTACATACACACCTA and reverse primer: CGTTGGTGCTGGGACTTTAT). The amplified PCR product was sequenced to confirm the genetic stability of the codon-pair-deoptimized gene.

#### 2.3.4. In Vitro Replication Kinetics of E38-ORF7 CPD

The capacities of the E38 original virus and E38-ORF7 CPD to replicate in vitro were examined by infecting PAM cells at a multiplicity of infection (MOI) of 0.1. Virus harvested from the infected cells and the titer of the virus was determined every day until the 5th day post-infection. Virus titration was performed at 0, 24, 48, 72, 96, and 120 h post-infection (hpi) using the infected cell culture medium as described in our previous work [11,38].

### 2.4. Animal Studies

#### 2.4.1. Attenuation of Virulence and Evaluation of Protective Immunity in Pigs

25 commercial cross-bred 3-week-old pigs, which were negative for PRRSV-specific antigen and antibodies, were divided into three groups to examine the change of virulence by CPD. In the first group, pigs were intranasally inoculated with 2 mL of 10^5^ TCID50/dose E38 original virus. In the second group, pigs were intranasally inoculated with 2 mL of 10^5^ TCID50/dose E38-ORF7 CPD. The negative group was inoculated with phosphate-buffered saline. All pigs were monitored for clinical respiratory signs and increased body temperature on a daily basis until 5 days post-inoculation (dpi). Weight gain, blood viral antigens, and serum antibodies were examined on a weekly basis. Five pigs in group 1 and 2 and 2 pigs in group 3 were necropsied at 14 dpi, while the other pigs were necropsied at 28 dpi. Tissues including lung were collected at necropsy for pathology and the viral copy number. The design of the animal experiment is presented Table 1.

#### 2.4.2. Evaluation of the Protective Immune Response Induced by E38-ORF7 CPD

Fourteen commercial cross-bred 3-week-old pigs were divided into four groups (Table 2). Four pigs received E38-ORF7 CPD intramuscularly, four pigs received E38-ORF7 CPD intradermally, four pigs were assigned to the unvaccinated and challenged group, and two pigs were assigned to the unvaccinated and unchallenged group. The first group intramuscularly received 2 mL of 10^4^ TCID50/dose E38-ORF7 CPD. The second group intradermally received 0.5 mL of 10^4^ TCID50/dose E38-ORF7 CPD. The other two groups (UnVac/Ch and UnVac/UnCh) were unvaccinated. The animals were challenged with 3ml of 10^5^ TCID50/mL CBNU0495 virus at 5 weeks post-vaccination (wpv), except for those in the unvaccinated and unchallenged group. Clinical signs were observed daily during the experimental period from post-vaccination to post-vaccination. Rectal temperature and body weight were recorded once a week. Serum and blood samples were taken for immunological assays and estimation of viral load. At autopsy (2 wpc), lung and lymph node samples were collected to observe gross and histopathological lesions.

#### 2.4.3. Serology

Serum samples were collected every week for serological assays. The serum samples were tested for the presence of PRRSV-specific antibodies using a commercial PRRSV ELISA kit (HerdCheck PRRS 3XR^TM^; IDEXX Laboratories Inc., Westbrook, ME, USA). Serum samples with signals higher than the antibody cut-off value (S/P ratio greater than 0.4) were considered positive.

#### 2.4.4. Quantification of Viral RNA

RNA was extracted from serum samples to quantify PRRSV genomic cDNA copy numbers, as previous described [38]. Real-time RT-PCR was designed to detect ORF7 sequences of viruses used in this study using primers (forward primer: GCCTTTAGCATTACATACACACCTA and reverse primer: CGTTGGTGCTGGGACTTTAT) for E38-ORF7 CPD and the unattenuated E38 original virus (forward primer: ATGGCCAGCCAGTCAATCA and reverse primer: TCGCCCTAATTGAATAGGTGA).

#### 2.4.5. ELISPOT (Enzyme-Linked Immunospot) Assay

PRRSV-specific IFN-γ secreting cells were quantified from a peripheral blood mononuclear cell (PBMC) layer of swine venous blood as described previously [39]. Venous blood mixed with Lymphoprep™ was centrifuged on a discontinuous gradient to collect PBMCs, and 5 × 10^5^ cells were seeded into each well of a porcine IFN-γ ELISPOT plate (MABTECH, Cincinnati, OH, USA) coated with a porcine IFN-γ capture antibody (10 µg/mL). PBMCs were then stimulated with E38 original virus at a MOI of 0.1 as the antigen. After 24 h incubation at 37 °C in a 5% CO_2_ atmosphere, cells were removed. Unstimulated and PHA-stimulated cells served as negative and positive controls, respectively. The number of PRRSV-specific IFN-γ secreting cells were expressed as responding cells in 10^6^ PBMCs.

#### 2.4.6. Pathology

At 2 wpc, all animals were euthanized and a collective autopsy was performed to examine gross pathological and microscopic histological lesions of the lungs and lymph nodes. Interstitial pneumonia was scored as follows: 0, no lesion (normal); 1, mild interstitial pneumonia; 2, moderate multifocal lesion; 3, moderate diffuse lesion; and 4, severe pneumonia.

### 2.5. Statistical Analysis

SPSS 16.0 version software was used to perform statistical analyses. The Mann–Whitney test was used to statistically compare the E38-ORF7 CPD and E38 original virus groups. Prior to analysis, the values were log transformed for viremia. The normality of the distribution of the examined variables was evaluated by the Shapiro–Wilk test. Continuous data were compared between groups using the Student’s t test followed by a one-way analysis of variance. Lung microscopic and respiratory scores were compared by the non-parametric Mann–Whitney test. All data are expressed as the mean ± standard deviation. For all statistical analyses, *p* < 0.05 was considered significant.

## 3. Results

### 3.1. Application of CPD to Generate E38-ORF7 CPD

Codon-pair-deoptimized attenuated E38-ORF7 CPD generated by CPD was successfully rescued from MARC-145 cells. The CPB value decreased to −0.2251 in the vaccine candidate strain (E38-ORF7 CPD) from −0.0371 in the E38 original (wild-type strain) (Table 3). A PRRSV-specific cytopathic effect identified by clumping and apoptosis of cells was observed in MARC-145 cells transfected with E38-ORF7 CPD. An immunofluorescence assay confirmed the replication of E38-ORF7 CPD.

To analyze the stability of each virus in cells and to determine the master seed passage threshold, successive passages were performed in MARC-145 cells. Analysis of the sequence of the entire ORF7 gene of the attenuated virus confirmed the presence of the mutation. The stability of the 22 base pair mutation in the ORF7 sequence artificially created for attenuation was confirmed.

The titer of rescued E38-ORF7 CPD peaked at 1 day after infection, similar to the E38 original virus. However, from 1 to 5 days after infection, the titer of E38-ORF7 CPD in PAM cells was 100 times lower than that of the E38 original virus having a statistically significant (*p* < 0.05) difference between the two viruses (Figure 2).

### 3.2. Attenuation of Virulence and Induction of Immune Response by CPD

Rectal temperature was monitored for 5 days post-inoculation. Rectal temperature of animals treated with the E38 original virus and E38-ORF7 CPD did not significantly differ from that of control animals. No animal had a temperature of 40.5 °C or higher, or showed signs of anorexia or any sort of distress (Figure 3A). Daily weight gain did not significantly differ between animals in the different groups (Figure 3B).

The number of viral antigens in blood was evaluated using RT-PCR. The number of viral genomes was 100 times lower in pigs inoculated with E38-ORF7 CPD than in pigs inoculated with the E38 original virus. The amount of total viral antigens in serum at 1 and 2 weeks after inoculation significantly differed between the two groups. No virus was detected in control pigs (Figure 3C).

Antibodies were measured using a commercial ELISA kit (IDEXX 3XR PRRSV Kit). At 2 weeks post-inoculation (wpi), PRRSV-specific antibodies were detected. Seroconversion was observed from 2 wpi and remained same until the end of the experiment. The antibody level did not significantly differ between pigs inoculated with E38-ORF7 CPD and those inoculated with the E38 original virus. No antibodies were identified in control pigs (Figure 3D).

An autopsy was performed, and lung lesion scores, histopathological lesions, and lung tissue viral loads were compared between the groups (Table 4 and Figure 4). At 2 wpi, pigs inoculated with the E38 original virus had significantly higher lung lesion scores and viral load than pigs inoculated with E38-ORF7 CPD and control pigs. E38 original inoculated pigs had severe, diffuse interstitial pneumonia. At 4 wpi, lung lesion scores did not significantly differ between pigs inoculated with E38-ORF7 CPD and control pigs. However, the PRRSV copy number was higher in the lung tissue of pigs inoculated with the E38 original virus than in lung tissue of pigs inoculated with E38-ORF7 CPD and control pigs.

### 3.3. Protective Immune Response Induced by E38-ORF7 CPD

The experimental animals were monitored daily for weight gain, temperature, and visible clinical changes from the day of vaccination until the end of experiment. Animals in the control group and those intramuscularly vaccinated with E38-ORF7 CPD were healthy based on their clinical appearance before challenge with the CBNU0495 virus. They exhibited anorexia for 1 dpc, but no animals in the vaccinated groups exhibited bleeding and congestion, joint edema, dyspnea, any other respiratory symptoms, or severe diarrhea. However, unvaccinated and challenged animals exhibited nasal discharge post-challenge. Average daily weight gain did not markedly differ between pigs in the 2 vaccinated groups, which was an average of 2.39 and 1.99 times higher in intradermally and intramuscularly vaccinated pigs than in unvaccinated and challenged pigs, respectively. Average daily weight gain did not significantly differ between the groups (Figure 5A).

The rectal temperature of unvaccinated and challenged animals was 41 °C and higher from 7 dpc, but did not significantly differ between the 2 vaccinated groups (Figure 5B). Animals vaccinated with E38-ORF7 CPD showed seroconversion from 14 dpi, and the S/P ratio gradually increased until 28 dpv. There was no significant difference in the antibody levels between the animals vaccinated with E38-ORF7 CPD intramuscular and intradermal route. The antibody titer in unvaccinated and challenged pigs was similar to that in vaccinated animals after 14 dpc.

The antibody titer in animals intramuscularly and intradermally vaccinated with E38-ORF7 CPD was similar until 35 days post-vaccination (dpv). However, it was slightly higher in intradermally vaccinated animals after 35 dpv (7 days post-challenge) than intramuscularly vaccinated animals, but this shows no statistically significant difference (Figure 5C).

PBMCs isolated from animals in the various groups were stimulated with virus to evaluate their ability to secrete IFN-γ. IFN-γ secreting cells were detectable in pigs immunized with E38-ORF7 CPD from 21 dpv onward and the level of these cells was significantly higher than the background. The level of IFN-γ secreting cells was higher in animals that received the vaccine intramuscularly than in those that received it intradermally. The level of IFN-γ secreting cells almost doubled by 28 dpv. The number of PRRSV-specific IFN-γ secreting cells remained at the basal level in unvaccinated and challenged, and unvaccinated and unchallenged animals (Figure 5D).

Following vaccination, viral genes of the E38-ORF7 CPD were detected and reached peaks at 14 dpv and gradually decreased until 28 dpv. The vaccinated challenged pigs exhibited lower levels of viremia in the serum compared to unvaccinated challenged pigs at 7 dpc, but this shows no statistically significant difference. In unvaccinated and unchallenged pigs, PRRSV was not detected in the serum throughout the experiment (Figure 5E).

At necropsy, the vaccinated challenged pigs had lower levels of viral load in lung tissue but this shows no statistically significant difference. PRRSV was not detected in the lung tissue of negative control pigs throughout the experiment dpc (Figure 5F).

### 3.4. Pathology

All experimental animals were euthanized and an autopsy was performed to examine gross pathological lesions of the lungs, tonsils, and lymph nodes. Lungs, lymph nodes, and tonsils of animals in the positive control group were characterized by hyperplasia. Red-to-purplish discoloration with multifocal hemorrhages was observed on the surface of the lungs of unvaccinated challenge animals. Animals in the other groups showed no visible lesions of the lungs, tonsils, and lymph nodes.

Animals in the unvaccinated and challenged group (positive control) showed severe interstitial pneumonia with a thickened alveolar septum, and accumulation of necrotic debris and inflammatory cells was evident in lung sections (Figure 6). Cell infiltration and hyperplastic lesions were similarly observed in the tonsils and lymph nodes. Animals that intramuscularly received E38-ORF7 CPD exhibited mild pneumonia and reduced lesions in the tonsils and lymph nodes compared with animals in the positive control group (Figure 6). Animals that intradermally received E38-ORF7 CPD showed lymph node hyperplasia on one side. Tissue hyperplasia was milder in animals that received E38-ORF7 CPD intramuscularly than in those that received it intradermally.

## 4. Discussion

PRRSV rapidly mutates during each replication cycle and has one of the highest evolutionary rates among RNA viruses [19,40,41]. This rapid mutation leads to continuous evolution of PRRSV and the generation of new variants, which makes it very difficult to develop a control strategy. MLVs are time-consuming to develop because they are generated by repetitive prolonged or long-term cell culture under a selective pressure, which incorporates random mutations [42]. The probability of MLVs reverting back to the virulent wild-type form or recombining with a circulating virus to form a highly pathogenic virus is high, leading to safety concerns [20,43,44]. Killed vaccines are safe to use but have efficacy issues due to their limited immunogenicity [45].

CPD is an advanced synthetic biology method that overcomes the drawbacks of traditional live attenuated vaccines. A vaccine is rapidly and easily designed using a computer algorithm by following simple steps that incorporate several synonymous mutations without altering the amino acid composition of the vaccine virus [20,24,46]. Several viruses including PRRSV have been attenuated using CPD technology. The NSP2, NSP9, ORF5, and NSP1 genes of PRRSV-2 have been successfully targeted to prepare codon-pair-deoptimized viruses [28,29,30].

It is questionable whether PRRSV-2 vaccines offer heterologous protection [45,47]. This study sought to develop a codon-pair-deoptimized PRRSV-1 vaccine that offers specific protection against locally circulating strains of PRRSV-1.

Previous studies performed CPD on PRRSV-2 via manipulating different genes by introducing the silent mutation of 78–95 nucleotides. In our previous study, the NSP1 gene of PRRSV-2 LMY and BP2017 was attenuated by CPD [11] and 86–91 nucleotides were manipulated without altering the amino acid sequence.

E38 is a PRRSV-1 isolated from a sow with reproductive failure on a Korean swine farm. We attempted to perform CPD of various genes. However, the attenuated virus could not be rescued from cell cultures. Finally, we successfully attenuated 22 nucleotides in ORF7 of PRRSV-1 and rescued the virus. ORF7 is a highly conserved gene encoding the N protein, which is an immunogenic protein that plays a vital role in the virus life cycle [48]. In the current study, CPD of ORF7 of E38 significantly attenuated the virus, with a significant reduction (100–1000 times) in the average replication capacity of the codon-pair-deoptimized virus compared with the E38 original virus. The stability of the 22 base pair mutation in ORF7 was confirmed. There was no mutation of the amino acid sequence, which indicates that the protein structure of the codon-pair-deoptimized virus will be similar to that of the original virulent virus at the ORF7 segment and that the codon-pair-deoptimized virus will induce an immune response similar to the original virulent virus [21,27].

In a previous study, the CPB values for attenuation of the LMY and BP2017 viruses were −0.2393 and −0.2238, respectively [11]. The optimal CPB value for attenuation of the ORF7 gene of E38 was −0.2371 in our study. The optimal CPB value to attenuate viral virulence varies depending on the selected genes and virus strain [11].

In vivo replication of E38-ORF7 CPD in pigs was reduced up to 100 times, with a reduced serum viral load, no clinical signs, and reduced or no pathological lesions in pigs inoculated with E38-ORF7 CPD compared with those inoculated with the E8 original virus. Serum antibody levels did not significantly differ between pigs inoculated with E38-ORF7 CPD and those inoculated with the E38 original virus. This indicates that the virus was successfully attenuated at the ORF7 gene, with retainment of immunogenicity despite reduction of its in vivo replication ability. These parameters are consistent with our previous study [11,31] and other similar studies [25,26,28,30,38].

E38-ORF7 CPD displayed a protective ability against the challenge virus by inducing immunity in pigs. The serum antibody titer was similar in animals that received the vaccine intramuscularly and those that received it intradermally. All vaccinated animals showed seroconversion by 14 days post-vaccination. This correlates with other studies, which also showed seroconversion by day 14 [30,31]. The humoral immune response was similar following intramuscular and intradermal inoculation. IFN-γ secreting cells were observed on day 21 and their level had doubled by day 28. Animals that received the vaccine intramuscularly had significantly more IFN-γ-secreting PBMCs than those that received it intradermally. On the contrary, in our previous study, the intradermally vaccinated group had higher numbers of IFN-γ secreting PBMCs than the intramuscularly vaccinated group. The type of virus isolate used may contribute to the difference in the immune response [49].

Gross and histopathological lung lesions are the main indicators of PRRSV infection [50]. Tissue hyperplasia and hemorrhage in the lungs are the typical characteristics of PRRSV infection and were evident in unvaccinated and challenged animals (positive control). Gross pathological lesions were significantly reduced in animals in both vaccinated groups at necropsy. Histopathological lung lesions did not significantly differ between intramuscularly and intradermally vaccinated animals, indicating that E38-ORF7 CPD can prevent gross tissue lesions caused by PRRSV challenge. Unvaccinated and challenged animals had severe interstitial pneumonia with a thickened alveolar septum, and accumulation of necrotic debris and inflammatory cells was evident in lung sections. Animals intramuscularly and intradermally vaccinated with E38-ORF7 CPD showed mild histopathological lesions in the lungs compared with the positive control group. Microscopic lesions in the tissues were slightly more severe in animals that received E38-ORF7 CPD intradermally than in those that received it intramuscularly. Based on IFN-γ stimulation and gross and histopathological examinations, pigs that received the vaccine intramuscularly were better protected than those that received it intradermally.

In the future, CPD of PRRSV-1 can be achieved by adding suitable adjuvants or T/B cell epitope coding nucleotides to the virus backbone. Along with vaccination, it is necessary to adopt strict biosecurity laws, implement good management practices, and perform regular monitoring of farms to eliminate PRRSV from swine farms.

## 5. Conclusions

PRRSV-1 subtype 1A is highly prevalent among the swine farms of Republic of Korea. PRRSV-2 codon-pair-deoptimized modified live virus vaccine does not give complete protection against PRRSV-1. Hence, we have developed a CPD vaccine targeting the ORF7 gene of subtype 1A of PRRSV-1. The vaccine protected the pigs by significantly reducing the virus replication and tissue lesions post-challenge with virulent virus, CBNU0495. Additionally, PRRSV-specific serum antibody and IFN-γ secreting cells increased from 14 dpv and 21 dpv onwards, respectively. This demonstrates that the CPD vaccine developed against PRRSV-1 is successful in protecting the susceptible host animals against the challenge virus CBNU0495.

## Figures and Tables

**Figure 1 vaccines-11-00777-f001:**
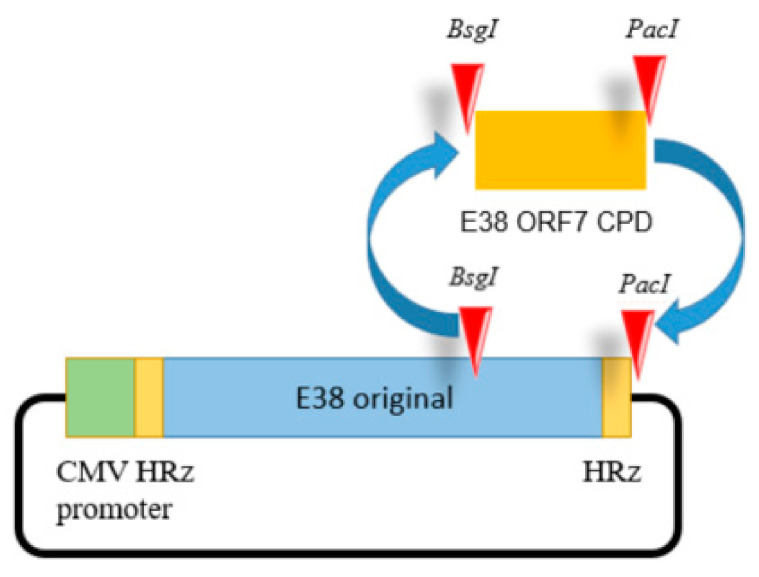
The E38-ORF7 CPD infectious clone containing the CMV promoter. The ORF7 fragment of the original E38 virus was replaced by the ORF7 attenuated fragment generated by CPD at the *BsgI* and *PacI* restriction sites. The presence of the ORF7 attenuated fragment was confirmed using PCR.

**Figure 2 vaccines-11-00777-f002:**
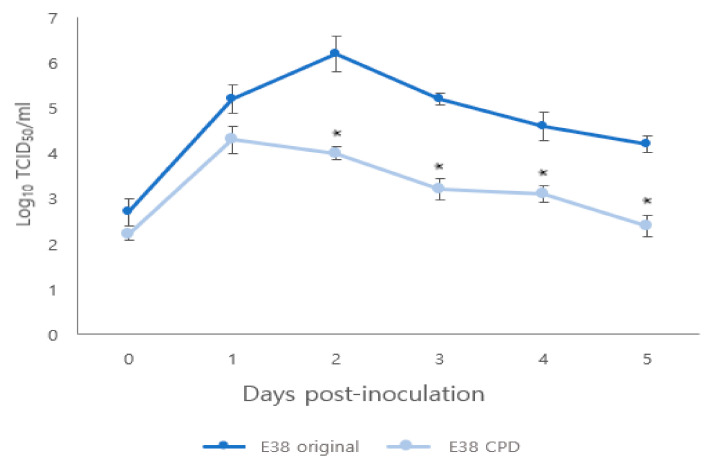
Comparison of replication kinetics of original and CPD viruses in PAM cells. The virus titers are presented as TCID_50_. Significant difference (*p* < 0.05) is indicated by ‘*’.

**Figure 3 vaccines-11-00777-f003:**
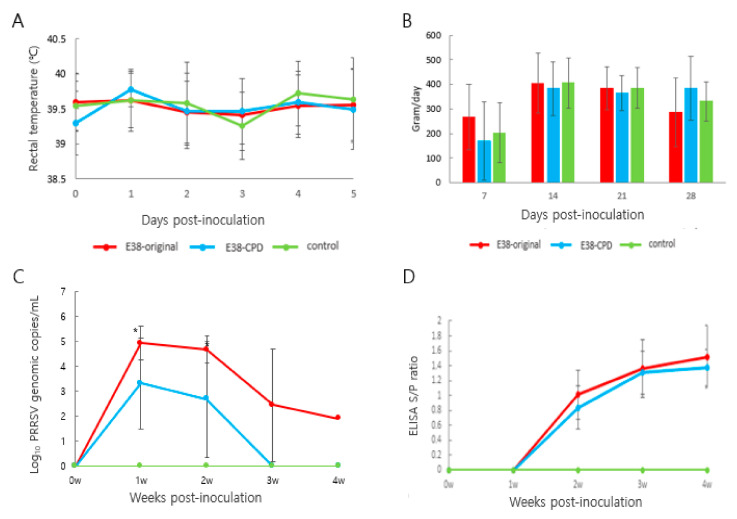
Assessment of the virulence and immune response induced by E38-ORF7 CPD and original virus: Three-week-old pigs were administration with 2 mL of 10^5^TCID50/dose attenuated virus (E38-ORF7 CPD) and original virus. The pigs were monitored for clinical signs, rectal temperatures, average daily weight gain, antibody levels, and viral copy number. (**A**) Rectal temperature (°C) of the pigs was monitored every day for 5 days. (**B**) Average Daily Weight Gain (ADWG) post-inoculation of the attenuated (E38-ORF7 CPD) and original virus (E38). There was no statistically significant difference between animal groups inoculated with the attenuated virus and or original virus and the control group. (**C**) Serum genomic copy numbers (PRRSV) in the animals inoculated with attenuated virus and or original virus. The statistically significant difference between the two groups is depicted as *, where *p* < 0.05. (**D**) SV-specific antibody level (ELISA S/P ratio) in the animals inoculated with attenuated and the original virus (E38).

**Figure 4 vaccines-11-00777-f004:**
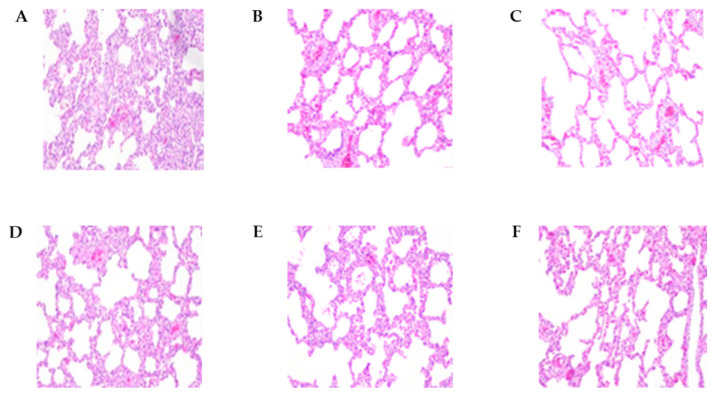
The histopathological lesions of necropsied pig lungs at 2 wpi and 4 wpi. (**A**) E38 original virus group lung lesions 2 weeks post-inoculation (wpi). (**B**) E38-ORF7 CPD group lung lesions 2 wpi. (**C**) Control group lung lesions 2 wpi. (**D**) E38 original virus group lung lesions 4 wpi. (**E**) E38-ORF7 CPD group lung lesions 4 wpi. (**F**) Control group lung lesions 4 wpi.

**Figure 5 vaccines-11-00777-f005:**
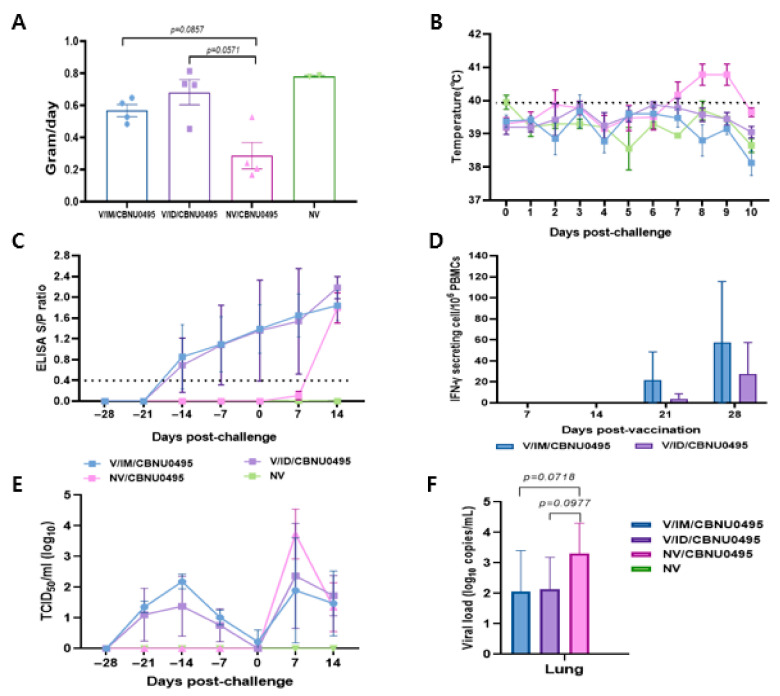
Induction of protective immunity by E38-ORF7 CPD. A group of 3-week-old pigs were vaccinated with E38-ORF7 CPD via intramuscular and intradermal routes. At 4 weeks post-vaccination animals were challenged with 3 mL of 10^5^TCID_50_/_mL_ heterologous virus (CBNU0495). Unvaccinated challenged and unchallenged groups were maintained as control groups. Clinical signs, rectal temperatures, average daily weight gain, serum antibody level, and IFN-γ were monitored. (**A**) Average daily weight gain of pigs observed during 0 dpc to 14 dpc (until the end of experiment). (**B**) Rectal temperature (°C) of the animals for 10 days post-challenge with the virulent PRRSV-1 virus. (**C**) The PRRSV-specific antibody titer measured by an indirect ELISA is expressed as the S/P ratio. (**D**) ELISPOT analysis showing the number of IFN-γ secreting PBMCs at 21 and 28 dpv. (**E**). Genomic copies of PRRSV in the serum samples. (**F**). PRRSV genomic copies in lungs of experimental animals at 14 dpc.

**Figure 6 vaccines-11-00777-f006:**
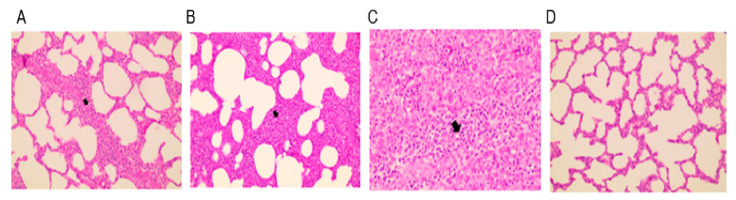
Histopathological lesions in the lungs of experimental animals. (**A**) E38-ORF7 CPD intramuscular group (V/IM/Challenge Virus): alveoli show mild hyperplasia and cell accumulation. (**B**) E38-ORF7 CPD intradermal group (V/ID/Challenge Virus): the alveolar septum is slightly thicker than in animals that received the vaccine intramuscularly. (**C**) UnVac/Ch (Positive control): the alveolar septum is thickened with accumulation of necrotic debris and inflammatory cells. (**D**) UnVac/UnCh (Negative control): alveoli and the alveolar septum are clear and normal. The differences were found to be statistically significant (*p* < 0.05) between the vaccinated group and the positive control (unvaccinated challenged group).

**Table 1 vaccines-11-00777-t001:** Attenuation of virulence by CPD.

Group	Inoculation	Total Number	1st Necropsy (14 dpi)	2nd Necropsy (28 dpi)
1	E38 original	10	5	5
2	E38-ORF7 CPD	10	5	5
3	Negative Control	5	3	2

**Table 2 vaccines-11-00777-t002:** Animal experiment to evaluate the protective efficacy of E38-ORF7 CPD.

Group	Inoculation	Dose	Total Number	Challenge (5 wpv)
1	E38-ORF7 CPD-IM route	10^4^ TCID_50_	4	CBNU0495
2	E38-ORF7 CPD-ID route	10^4^ TCID_50_	4	CBNU0495
3	Positive Control (UnVac/Ch)	-	4	CBNU0495
4	Negative Control (UnVac/UnCh)	-	2	-

**Table 3 vaccines-11-00777-t003:** Comparison of the E38 original virus and E38-ORF7 CPD.

Virus	Number of Silent Mutations (nt)	CpG ^a^	UpA ^b^	CPB ^c^
E38 original	-	0.5714	0.2610	−0.0371
E38-ORF7 CPD	22	1.0797	0.6185	−0.2251

^a^ Codon pair bias (CPB) of the corresponding sequence. ^b^ Ratio of observed dinucleotide frequency to that expected based on mononucleotide composition. F(CpG)/F(C) × F(G). ^c^ Ratio of observed dinucleotide frequency to that expected based on mononucleotide composition. F(UpA)/F(U) × F(A).3.2. Replication Capacity of E38-ORF7 CPD in PAM Cells

**Table 4 vaccines-11-00777-t004:** Summary of the results regarding gross lung lesions and viral load in lungs.

	2 wpi			4 wpi		
	E38 Original	E38-ORF7 CPD	Control	E38 Original	E38-ORF7 CPD	Control
Lung lesion score	1.8 ± 0.44 *	0.53 ± 0.6	0.11 ± 0.19	0.53 ± 0.38	0.33 ± 0.33	0.16 ± 0.23
Log_10_ PRRSV genomic copies/g	5.59 ± 1.02 *	1.72 ± 1.69	0	2.3 ± 3.19	0.76 ± 1.69	0

* indicates significant difference (*p* < 0.05) between E38 original virus and E38-ORF7 CPD virus in virulence factor.

## Data Availability

The datasets generated or analyzed during this study are available from the corresponding author on reasonable request.

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
