# Peer review of "Codon Pair Deoptimization (CPD)-Attenuated PRRSV-1 Vaccination Exhibit Immunity to Virulent PRRSV Challenge in Pigs"

_vaccines, 2023, doi:10.3390/vaccines11040777_

Round 1

Reviewer 1 Report

The development of new vaccines for porcine reproductive respiratory syndrome is of great significance for the pig industry. In this manuscript, codon pair deoptimization (CPD) is used to get a vaccine candidate to overcomes the disadvantages of MLV vaccines and the efficacy and safety of the vaccine was tested with good protect against virulent PRRSV-1.

1.Please reorganize the introduction and clarify the reasons for reserch on this kind of vaccine.

2.It is suggested to express the contents of Table 1, 2 and 3 in text. Also, recommended that the histopathological examination in table 4 be mapped separately.

3. Please add the data of gross lesions of the experimental animals.

4. Please add the data of viral load in tissues of the infected animals after vaccination. In addition, please add the results of immunohistochemistry test in Figure 5, which can effectively explain the efficacy of the vaccine.

5. Please reorganize the discuusion. Please explain in detail the advantages of the developed new vaccine compared with the commercialized vaccine.

Author Response

  1. Please reorganize the introduction and clarify the reasons for research on this kind of vaccine

≫ We corrected as your comment.

  1. It is suggested to express the contents of table 1, 2 and 3 in text. Also, recommended that the histopathological examination in table 4 may be mapped separately.

≫ We corrected as your comment.

Table 1: Three groups of twenty-five commercial cross bred 3-week-old pigs were formed. The effect of attenuation was studied, as well as the protective efficacy of attenuated and unattenuated viruses. As shown in the table, necropsies were performed in two batches, the first at 14 days post inoculation (dpi) and the second at 28 dpi

Table 2: Four groups of fourteen commercial cross-bred 3-week old pigs were formed. Groups 1 and 2 received 2 ml of 4 TCID50 of vaccine (E38-ORF7 CPD) via intramuscular and intradermal routes respectively. The other two groups, positive and negative controls, were not immunized. The vaccinated and unvaccinated positive control groups were challenged 5 weeks after vaccination (wpv) with a 105 TCID50/ml dose of challenge virus (CBNU0495) and necropsy was performed on all animals two weeks later (7wpv or 2 wpc)

Table 3: The number of silent mutations introduced in the E38 original virus (unattenuated) and the E38-ORF7 CPD (attenuated) were compared, with no silent mutations observed in the E38 original virus (unattenuated), but silent mutations of 22 nucleotides observed in the ORF7 region in the E38-ORF7 CPD. When compared to the unattenuated E38 original virus, the CPB ratio in E38-ORF7 CPD was -0.2251.

  1. Please add the data of gross lesions of the experimental animals

≫ The manuscript included histopathological lesions in lung. We removed the description of gross as your comment.

  1. Please add the data of viral load in tissues of the infected animals after vaccination. in addition, please add the results of immunohistochemistry test in figure 5, which can effectively explain the efficacy of the vaccine
    ≫ We corrected as your comment.

  2. Please re-organize the discussion, please explain in detail the advantages of the developed new vaccine compared with the commercialized vaccine.

≫ We corrected as your comment.

The immune response of the attenuated virus was compared to that of the unattenuated virus in this study. The CPD vaccine is made by inserting synonymous codons into the target gene. When the virus multiplies in the cells, the rate of mutation is much lower than in commercial vaccine modified live virus. In general, the chances of reversing virulence of CPD vaccine is far less than those of commercial vaccine live virus. This is the advantage of the new vaccine over commercialized vaccine. In this study, we did not compare the attenuated CPD vaccine to the commercialized vaccine.

Reviewer 2 Report

Dear Authors, Congrats for your work, very interesting study. It looks like a promising technology.  

IN ABSTRACT: MLV vaccines do not fail to prevent infection. They cannot prevent infection so they do not fail because practically there is no vaccine providing sterilizing immunity. So, it is not a failure, it is a limitation. I would recommend to change the word “fail”. One thing is to rever to virulent and other recombination. Recombiantion is not a reversion to virulence because it becomes a new entity, most of the times it is the main wild virus with some small elements of the vaccine virus (usullually the ORF 5  in betaarterivirus suid 1, so that is how it is detected.

LINE 31. “Piglets” makes reference a suckling piglets. I recommend to use the word “pigs”.

LINE 56: This sentence is not correct – and it looks like deliberately incorrect -  as cross protection is partial but happens between subtypes and even between type 1 and 2. It is true that the partial cross-protection is not enough to protect against clinical disease, but we are talking about a disease that not even homologous protection is able to prevent infection or clinical disease. Therefore is not right to say that vaccines systematically fail or that cross-protection does not exist cause it does. It is the virus who achieves to avoid and misdirect the immune response.

LINE 65: Please, explain how you can state that the new virus lacks virulence if it is able to multiple in macrophages as the other MLV, and when it multiplies in the macrophages it leads to some degree of pathological effect. This statement is not referenced and it is not correct to put it in the introduction if this is the conclusion of your study. Additionally, on your study, your CPD replicated and caused some degree of lesions as on Table B so it did not lack virulence.

LINE 75. Predominance of type 2 just applies to certain territories (please clarify, I guess you may add it is just Korea or Asia). In Europe, type 1 is the mostly predominant.

LINE 305. “ impotence” please review if the meaning on this word. It is not frequently used to describe the lack of energy, weakness or apathy on pigs.

FIGURE 4. Please increase resolution. It is too blurry to be read. Please, checl the legends as you use different names for the treatments groups.

FIGURE 5: The number of strain can be changed by “challenged” as that is the term you used to denominate that group.

LINE 370: Typo on RRSV

LINE 375: It is really debatable that MLV only includes a “few” mutations. Certainly they are random, but mutations are a lot as the virus mutates very quickly and the number of passages is high. Additionally culture are done in non-porcine cell, which can produce some selection and fix certain mutations. Please, reconsider if “few” is adequate and explain.

LINE 376: In Europe, most MLV vaccines are considered safe and they are described in that way; there are a few known cases of reversion affecting a few products, but the main products are considered safe. Please, reconsider if the reference you use really supports the statement you make, and in that case explain it please. The part is leading to safety concerns in fine.

LINE 381: The very process of replicating the virus in a non-procine culture to create the vaccine product is similar to the MLV, therefore, you also introduce some random mutations similar to MLV (certainly it may occur in a lower degree). Therefore, the immunogenicity is going to be altered as in MLV, probably in a lower degree, but it is not black vs white as you state in your sentence “without altering immunogenicity” especially when you alter the nucleocapsid genome. Additionally, you do not know if the mutations you introduced have or not an effect on immunogenicity.    

LINE 398. The sentence of “the majority …” looks like orphan. Are you talking about your study or the literature? Please, confirm it is DNA vaccination.

LINE 401. The comparison was never made against the E38 original virus happened in vitro, so it may be added to the statement. You have not presented data in vivo.

LINE 402. That is just a theory. As the CPD needs to be replicated to be transformed into a vaccine product, mutations can occur, in that or in other parts on the genome. Additionally, your study was restricted to that segments and disregarded possible changes on the genome sequence on other areas of the viral genome. Consequently, your statement must be restricted to the segment of the virus you studied which only represent and small part of the total genome.

LINE 411. You compare the vaccine virus vs an wild-virus of unknown virulence, not even the original E38.

LINE 416; The loss of pathogenicity could be quite incorrect as Table 4 clearly show some degree of damage. The group E38 may be significantly different but what about differences in between the the group control and the group with E38 CPD? Additionally, viral load means replication and replication means macrophage loss so immunosuppression. This effect could be dose dependant, and the fact that it created some immunity for the challenge you did on seronegative animals, though no light at all because protection must be evaluated in seropositive animals too as in real condition, seronegative piglets are rarely vaccinated for PRRSV, the real challenge for vaccines is to overcome the MDI. Therefore, it looks like the vaccine strain pathogenicity is lower than the parental virus but further studies are needed to be able to claim a loss of pathogenicity.

CONCLUSIONS: Same as before, it is only one challenge in very specific conditions with just one strain (you do not provide any information of similarity in between the challenge strain and E38) on naïve animals (no MDA interference) so it is fair to claim you succeed in protect in your challenge, but it is to much to escalate the claim to all the PRRSv1 in the Korean pig industry.   

Author Response

  1. In abstract: MLV vaccines do not fail to prevent infection. They cannot prevent infection so they do not fail because practically there is no vaccine providing sterilizing immunity. So, its not a failure, it is a limitation. I would recommend to change the word “fail’. One thing is to rever to virulent and other recombination. Recombination is not a reversion to virulence because it becomes a new entity, most of the times it is the main wild virus with some small elements of the vaccine virus (usually the ORF5 in betaarterivirus suid 1 so that is how it is detected

            ≫ We corrected manuscript as your comment.

  1. Line 31: ‘piglets’ makes reference a suckling piglet. I would recommend to use the word ‘Pigs’

≫ We changed the word ‘piglets’ to ‘pigs’ throughout the manuscript as your comment.

  1. Line 56: this sentence is not correct-and it looks like deliberately incorrect- as cross protection is partial but happen between subtypes and even between type 1 and 2. It is true that the partial cross-protection is not enough to protect against clinical disease, but we are talking about a disease that not even homologous protection is able to prevent infection or clinical disease. Therefore, is not right to say that vaccines systematically fail or that cross-protection does not exist cause it does. It is the virus who achieves to avoid and misdirect the immune response.

≫ Alterations were made and included partial protection against heterologous virus.

  1. Line 65: Please, explain how you can state that the new virus lack virulence if it is able to multiple in macrophages as the other MLV, and when it multiplied in the macrophages it leads to some degree of pathogenic effect. This statement is not referenced and is not correct to put it in the introduction if this is the conclusion of your study. additionally, on your study, your CPD replicated and caused some degree of lesions as on Table B so it did not lack virulence.

≫ We removed the words lack of virulence.

  1. Line 75: predominance of type 2 just applies to certain territories (please clarify, I guess you may add it is just Korea or Asia). In Europe, type 1 is the mostly predominant.

≫ In Korea, PRRSV-2 was isolated in 1993, PRRSV-1 was isolated in 2005. Therefore, the PRRSV-2 MLV was licensed in 1995 and started to be used first. Recently, PRRSV-1 outbreaks have increased and the two genotypes are occurring similarly. Korea (territory) is now mentioned in the sentence.

  1. Line 305: “impotence” please review if the meaning on this word. It is not frequently used to describe the lack of energy, weakness or apathy on pigs.

≫ We removed the word ‘Impotence’ as your comment.

  1. Figure 4: Increase resolution.

≫ We corrected as your comment.

  1. Figure 5: number of strain can be changed by challenged as that is the term you used to dominate that group.

≫ We corrected as your comment.

  1. Line 305: Typo on RRSV

≫ We corrected ‘RRSV’ to ‘PRRSV’ as your comment.

  1. Line 375: it is really debatable that MLV only includes a few mutations. Certainly, they are random, but mutations are a lot as the virus mutates very quickly and the number of passages is high. Additionally, culture is done in non-porcine cell, which can produce some selection and fix certain mutations. Please, reconsider if “few’ is adequate and explain.

≫ We omitted the word “few” as your comment.

  1. Line 376: In Europe, most MLV vaccines are considered safe and they are described in that way; there are a few known cases of reversion affecting a few products, but the main products are considered safe. Please, consider if the reference you use really supports the statement you make, and in that case explain it please. The part is leading to safety concerns in fine.

≫ There are several manuscripts on the safety of PRRS live attenuated vaccine. Refer to the paper (Zhou et al., 2021, Vaccines, 9(4), 362.). This reference (Zhou et al., 2021) is included in the manuscript.

  1. Line 381: the very process of replicating the virus in a non-porcine culture to create to vaccine product is similar to the MLV, therefore, you can also introduce some random mutations similar to MLV (certainly it may occur in a lower degree). Therefore, the immunogenicity is going to be altered as in MLV, probably in a lower degree, but it is not black vs white as you state in your sentence ‘without altering immunogenicity’ especially when you alter the nucleocapsid genome. Additionally, you do not know if the mutations you introduced have or not an effect on immunogenicity.

≫ Codon pair deoptimization incorporates synonymous codons, resulting in the same amino acid coded. As a result, the resulting antigen or immunogen at the attenuated virus’s deoptimized region is similar to the virus before attenuation. we replaced immunogenicity with amino acid composition and provided the appropriate references.

  1. Line 398: the sentence of the majority looks like orphan. Are you talking about your study or the literature? Please, confirm it is DNA vaccination.

≫ It is from the literature and not from our research. We deleted the sentence.

  1. Line 401: the comparison was never made against the E38 original virus happened in vitro, so it may be added to the statement. You have not presented the data in vivo.

≫ The term ‘original virus’ refers to a virus that has not been attenuated. We compared CPD attenuated E38 (E38-ORF7 CPD) to unattenuated virus (E38-Original).

  1. Line 402: that is just a theory. As per CPD needs to be replicated to be transformed into a vaccine product, mutations can occur, in that or in other parts on the genome. Additionally, your study was restricted to that segments and disregarded possible changes on the genome sequence on other areas of the viral genome. Consequently, your statement must be restricted to the segment of the virus you studied which only represent and small part of the total genome.

≫ We corrected as your comment. The sequence ORF7 has been incorporated into the sentence.

  1. Line 411: you compare the vaccine virus vs a wild-virus of unknown virulence, not even the original E38.

≫ As a vaccine strain, it should be adapted to MARC-145 cells, and E38 was selected because it has better proliferation ability than other PRRSV-1. The PRRSV-1 is known to be less pathogenic that the PRRSV-2, and the E38 original virus was not suitable as a challenge virus because it showed mild clinical symptoms. Therefore, the most highly pathogenic PRRSV-1 was used. In Korea, only PRRSV-1 subtype1 occurs, of which subgroup 1A is more than 90%, and this challenge virus belongs to subtype 1A (E38 original virus) also belongs to subtype 1(subgroup A), so we believe it is representative as a challenge virus.

  1. Line 416: The loss of pathogenicity could be quite incorrect as table 4 clearly shows some degree of damage. The group E38 may be significantly different but what about the differences in between the two-group control and the group E38 CPD? Additionally, viral load means replication and replication means macrophage loss so immunosuppression. This effect could be dose dependent, and the fact that it created some immunity for the challenge you did on seronegative animals, though no light at all because protection must be evaluated in seropositive animals too in real conditions, seronegative piglets are rarely vaccinated for PRRSV, the real challenge for vaccines is to overcome the MDI. Therefore, it looks like the vaccines strain pathogenicity is lower than the parental virus but further studies are needed to be ae to claim a loss of pathogenicity.

≫ Pre-clinical evaluation, such as vaccine efficacy are usually performed in negative pigs. We plan to evaluate the vaccine in swine farm (under real condition) and will analysis the results considering PRRS infection, vaccination and MDA, etc.

  1. Conclusions: correction added.

≫ We corrected as your comment.

Similarity between the challenge virus and E38 is mentioned in the material and methods part as: nucleotide similarity between E38 (strain used for attenuation) and CBNU0495 (experimental challenge virus) is 93%.

Round 2

Reviewer 1 Report

All my concerns have been modified.